# The Discovery of New Inhibitors of Insulin-Regulated Aminopeptidase by a High-Throughput Screening of 400,000 Drug-like Compounds

**DOI:** 10.3390/ijms25074084

**Published:** 2024-04-06

**Authors:** Johan Gising, Saman Honarnejad, Maaike Bras, Gemma L. Baillie, Stuart P. McElroy, Philip S. Jones, Angus Morrison, Julia Beveridge, Mathias Hallberg, Mats Larhed

**Affiliations:** 1The Beijer Laboratory, Science for Life Laboratory, Department of Medicinal Chemistry, Biomedical Centre, Uppsala University, P.O. Box 574, SE-751 23 Uppsala, Sweden; jgbeveridge@gmail.com (J.B.); mats.larhed@ilk.uu.se (M.L.); 2Pivot Park Screening Centre, Kloosterstraat 9, 5349 AB Oss, The Netherlands; saman.honarnejad@ppscreeningcentre.com (S.H.); maiky103@hotmail.com (M.B.); 3BioAscent Discovery Ltd., Bo‘Ness Road, Newhouse, Motherwell ML1 5UH, UK; gbaillie@bioascent.com (G.L.B.); smcelroy@bioascent.com (S.P.M.); pjones@bioascent.com (P.S.J.); amorrison@bioascent.com (A.M.); 4The Beijer Laboratory, Department of Pharmaceutical Biosciences, Neuropharmacology and Addiction Research, Biomedical Centre, Uppsala University, P.O. Box 591, SE-751 24 Uppsala, Sweden; mathias.hallberg@uu.se

**Keywords:** insulin-regulated aminopeptidase (IRAP), ultra-high-throughput screening (uHTS)

## Abstract

With the ambition to identify novel chemical starting points that can be further optimized into small drug-like inhibitors of insulin-regulated aminopeptidase (IRAP) and serve as potential future cognitive enhancers in the clinic, we conducted an ultra-high-throughput screening campaign of a chemically diverse compound library of approximately 400,000 drug-like small molecules. Three biochemical and one biophysical assays were developed to enable large-scale screening and hit triaging. The screening funnel, designed to be compatible with high-density microplates, was established with two enzyme inhibition assays employing either fluorescent or absorbance readouts. As IRAP is a zinc-dependent enzyme, the remaining active compounds were further evaluated in the primary assay, albeit with the addition of zinc ions. Rescreening with zinc confirmed the inhibitory activity for most compounds, emphasizing a zinc-independent mechanism of action. Additionally, target engagement was confirmed using a complementary biophysical thermal shift assay where compounds causing positive/negative thermal shifts were considered genuine binders. Triaging based on biochemical activity, target engagement, and drug-likeness resulted in the selection of 50 qualified hits, of which the IC_50_ of 32 compounds was below 3.5 µM. Despite hydroxamic acid dominance, diverse chemotypes with biochemical activity and target engagement were discovered, including non-hydroxamic acid compounds. The most potent compound (QHL1) was resynthesized with a confirmed inhibitory IC_50_ of 320 nM. Amongst these compounds, 20 new compound structure classes were identified, providing many new starting points for the development of unique IRAP inhibitors. Detailed characterization and optimization of lead compounds, considering both hydroxamic acids and other diverse structures, are in progress for further exploration.

## 1. Introduction

It was demonstrated 35 years ago by Braszko et al. that intracerebroventricular injection of the endogenous hexapeptide angiotensin IV (Ang IV, Val-Tyr-Ile-His-Pro-Phe), which is formed after degradation of the hypertensive octapeptide angiotensin II (Ang II, Asp-Arg-Val-Tyr-Ile-His-Pro-Phe), improved memory and learning in rats [1]. The hexapeptide Ang IV exerted effects on both passive and conditioned avoidance responses and on motor activity. Subsequently, the hexapeptide was examined in a variety of experimental models [2,3,4,5,6,7]. Moreover, the effects of structural analogs of Ang IV were also explored, e.g., Nle-Ang IV (Nle-Tyr-Ile-His-Pro-Phe) and LVV-hemorphin-7 (Leu-Val-Val-Tyr-Pro-Trp-Thr-Gln-Arg-Phe), and these two peptides were shown to act as strong promoters of memory retention and retrieval in rats [4,7,8]. The binding sites of Ang IV in brain areas associated with cognition were early identified [9], e.g., in the hippocampus [10], and today, excellent reviews on Ang IV and its impact on cognition are available [11,12,13,14,15].

In 2001, Albiston and Chai presented data suggesting that insulin-regulated aminopeptidase (IRAP, oxytocinase, EC 3.4.11.3), which, alongside aminopeptidases ERAP1 and ERAP2, belongs to the oxytocinase subfamily of M1 zinc aminopeptidases, is a main target for Ang IV and that the hexapeptide exerts its actions through inhibition of this peptidase [16]. The membrane-bound zinc-metallopeptidase is highly expressed in brain regions associated with cognition [17]. IRAP was first identified as the enzyme that degrades oxytocin in the late stages of pregnancy and childbirth [18] and was demonstrated to process peptide hormones as vasopressin and other presumed in vivo substrates [19,20,21] known to influence cognitive functions favorably [12,22,23,24]. It has been suggested that the inhibition of IRAP may facilitate memory by increasing hippocampal dendritic spine density via a GLUT4-mediated mechanism [25].

The crystal structure of human IRAP has been determined and revealed a closed, four-domain arrangement with a large, mostly buried cavity adjacent to the active site. The crystal structure reveals that the GAMEN exopeptidase loop adopts a very different conformation from other aminopeptidases, which explains the specificity of IRAP for cyclic peptides such as oxytocin and vasopressin [26]. Various conformational changes in IRAP induced by ligands are reported [27,28]. Trafficking motifs at the N-terminal domain of IRAP regulate the movement of enzymes within different intracellular compartments [29].

Even though most focus has been on identifying drug-like molecules with potential as cognitive enhancers [12,30,31], e.g., for future treatment of Alzheimer’s disease, the important roles of IRAP in antigen cross-presentation [32] and T-cell receptor signaling [33], as well as neuroprotection in ischemic stroke model [34], has also attracted attention in drug discovery contexts.

In the first series of IRAP inhibitors derived from the endogenous hexapeptide Ang IV, various amino acid residues were replaced by unnatural residues often to improve stability [35]. Notably, a β-homoamino acid scan of Ang IV resulted in the identification of the metabolically stable derivative **AL-11** with a five-fold higher affinity to IRAP than Ang IV (*K*_i_ = 7.6 vs. 62 nM for Ang IV) in the assay applied [36]. Furthermore, replacements with conformationally constrained residues could deliver ligands, e.g., **AL-40**, (**IVDE77**, Appendix A) with >30-fold higher affinity (*K*_i_ = 1.7 nM) than Ang IV and improved metabolic stability [36,37,38]. Macrocyclic analogs of Ang IV, such as **HA08** (Appendix A), demonstrated a high inhibitory capacity of IRAP [39], enhanced spine densities, and reversed hydrogen peroxide-induced toxicity in primary hippocampal neuron cultures [40,41].

Although modified peptides, such as **AL-11**, **AL-40** (**IVDE77**), and **HA08**, are powerful IRAP inhibitors and useful research tools, they are not attractive as drug candidates and non-peptidic small molecules are therefore highly desired. The first series of drug-like small molecule inhibitors of IRAP were reported by Siew Chai and her group in Australia in 2008 [42]. An in silico screen of a library of approximately two million compounds using a homology model of IRAP, i.e., using the crystal structure of Leukotriene-A4 hydrolase [43], resulted in a lead series of IRAP inhibitors comprising a benzopyran scaffold. Some of the more potent compounds they developed were the racemic pyridine derivative **HFI-419** and the quinoline derivatives **HFI-435** and **HFI-437** with *K*_i_ values of 420, 360, and 20 nM, respectively. The benzopyran-based IRAP inhibitor **HFI-419** was shown to have high selectivity versus other aminopeptidases, such as APN, ERAP1, ERAP2, and LTA4H, and exerted a cognitive-enhancing effect in rodents after intracerebroventricular administration, similar to Ang IV and LVV-hemorphin-7 [7,42].

More recently, our group screened a library of 10,500 small molecules and identified various lead compounds [44], including arylsulfonamides [45], that enhance dendritic spine density in primary hippocampal neuron cultures [46], and un-competitive IRAP inhibitors encompassing a spiro-oxindole dihydroquinazolinone scaffold [47] and an imidazo[1,5-α]pyridine scaffold [48]. Additional selective inhibitor leads targeting an allosteric site in IRAP were identified after applying a virtual docking approach [49], and it was reported that modulators of the structurally related hERAP2 have also been identified from a high-throughput screen (HTS) [50].

Notably, very recently, it was reported by Stratikos that one of our synthesized aryl sulfonamides and **HFI-419** target a previously unidentified proximal allosteric site and utilize non-competitive inhibition mechanisms [51].

Several studies have used structure-based design to identify IRAP inhibitors including diaminobenzoic acids [52], pseudophosphinic peptides [53], and, more recently, selective inhibitors derived from amino acid derivatives of bestatin [54]. Several recent reviews describe the medicinal chemistry of the various IRAP inhibitors in detail [55,56,57]. To conclude, despite recent progress in identifying non-peptidic IRAP inhibitors, new small molecules that demonstrate isoform selectivity, good oral bioavailability, and can pass the Blood–Brain Barrier are highly desirable.

The prevailing strategy for lead generation utilized in successful hit-to-clinical campaigns predominantly originates from known starting points (59%) followed by random screening (21%) [58]. HTS is an attractive strategy to identify new chemical starting points, though the capability to conduct extensive screening is coupled with considerable infrastructural resources and access to a high-quality large compound library. The capability to perform such screening campaigns has predominantly been within large pharma companies. The European Lead Factory (ELF) is a collaborative consortium consisting of both industry and academic partners. Its core objective is to broaden access to this essential technique for drug discovery [59].

We herein report on the performance and output of an ultra-HTS of ~400,000 small molecules, the majority being from several large pharma companies in collaboration with the European Lead Factory (ELF). The compound collection encompasses contributions from various companies with diverse therapeutic area backgrounds, alongside entirely novel compounds synthesized within the library. This generates structural diversity and integrates molecules with complementary physicochemical properties, rendering the collection highly diverse and drug-like [60]. The screen was performed with the soluble truncated IRAP enzyme and used in a screening cascade consisting of three biochemical assays, developed in 1536 plate format, and an orthogonal thermal shift assay. This resulted in a qualified hit list of the most promising 50 compounds, the most potent of which has a pIC50 of 6.5 (IC50 of 0.32 µM).

## 2. Results

### 2.1. Development of the Screening Assay Cascade

#### 2.1.1. Reference Compounds

In the assay development, **HA08** [39], **IVDE77** [38], and **I** [45] were used as reference compounds (see Appendix A).

#### 2.1.2. Principle, Optimization, and Miniaturization of Primary and Orthogonal Assays

The primary biochemical assay was developed to measure the enzymatic activity of IRAP. The assay principle was based on a fluorogenic L-leucine-7-amido-4-methylcoumarin substrate, which, upon enzymatic cleavage by IRAP, resulted in the production of L-Leucine and the release of the fluorescent 7-amido-4-methylcoumarin group (Ex. 355 nm/Em. 460 nm) (see Appendix A). To establish an orthogonal assay, an alternative L-Leucine para-nitroanilide substrate was used, which IRAP turns over to L-Leucine and para-nitroaniline and is measurable by absorbance at 405 nm (Appendix A).

To ensure biochemically sound, optimal, robust, and affordable assay conditions for large-scale screening, several factors need to be considered. These include the final concentration of substrates, based on the determined substrate *K*_m_ value; the assay kinetics and reaction linearity; achieving a minimal, rate limiting protein concentration; identifying optimal buffer additives and concentrations; the choice of assay plate type; the final assay volume per well; the stability of reagents; the magnitude and variance of the signal; and the potency and rank order of reference compounds. For both primary fluorescence and orthogonal absorbance assays, non-binding 1536-well assay plates were used, and 0.05% BSA was included in the assay buffer to minimize the issues associated with protein sticking to plastics. The substrate *K*_m_ values were determined in two separate experiments using two different batches of IRAP protein. For this, the IRAP progress curve was monitored every 5 min for 2 h using an Envision multimode plate reader (PerkinElmer, Waltham, MA, USA). Using the steady-state (linear) enzyme velocity to calculate the reaction rate, the *K*_m_ for L-Leucine-7-amido-4-methylcoumarin was calculated as 36.6 μM (Appendix A), and for L-Leucine p-Nitroanilide, it was 0.38 mM (Appendix A). To evaluate if using substrate concentrations close to the determined *K*_m_ values results in sufficient dynamic range within the linear range of the enzymatic reaction, Z′ and S/B were calculated for multiple time points up to 2.5 h in the presence and absence of reference compounds in both assays. The primary fluorescence assay reached an optimal Z′-value after 1 h when using 37.5 μM substrate (Appendix A). For the absorbance, 2 h was required for the enzyme reaction time; however, using ~0.4 mM substrate was associated with increased assay variation and a poor Z′ (Appendix A). As such, 1 mM substrate concentration was used for the final assay since the potency of the reference compounds remained comparable yet the assay robustness improved (Appendix A).

To minimize the amount of reagent required for the screening campaign and evaluate reagent stability, the primary assay performance and response to reference compounds were assessed in a 4 μL final volume using freshly prepared reagents vs. reagents stored overnight at 4 °C or RT (Appendix A). For reagents stored overnight at RT or 4 °C, the maximum counts, and, therefore, the S/B ratios, were slightly decreased; however, the Z′-values remained excellent (~0.9). The IC_50_ values for all reference compounds were comparable to the IC_50_ values generated when using an 8 μL final volume (Appendix A). Together these data indicate that the lower volume assay is fit for purpose for a large-scale, multi-day HTS campaign.

Finally, to evaluate assay performance in a fully automated fashion and assess assay/target liabilities toward compounds with undesired modes of action, the final primary assay protocol was performed in four independent 1536-well assay plates. Two plates were used as “QC plates” to assess full-plate signal variability and quality, including reference compound potency, and two plates were used to profile a robustness set (RS) library that contains known assay interference compounds alongside compounds with no known problematic activity, so-called “clean” compounds [61]. For all plates, S/B ratios and Z′-values were comparable, the %CV was acceptable (<5%), and the responses to reference compounds were reproducible. For plates used to profile the RS, the S/B ratio (~30) and Z′-value (~0.9) were good for both plates (Appendix A). Very few RS compounds showed strong inhibition of the assay, indicating a low propensity for promiscuous nuisance compounds. With a cut-off of at Z-score ≤ −4, 15 compounds were flagged as inhibitors, which were mainly clean compounds (three aggregators, two chromophores, one fluorophore, one luciferase inhibitor, and eight clean) (Appendix A).

The Thermal shift assay (TSA) is a biophysical technique measuring protein thermal stability. The assay relies on a fluorescent dye that is quenched in an aqueous environment but fluoresces when de-solvated in non-polar environments, such as the hydrophobic pockets of proteins. The target protein is mixed with the dye and heated gradually until it becomes completely denatured; as it unfolds or melts, the dye binds to newly exposed hydrophobic regions of the protein, resulting in a significant increase in fluorescence emission detected by a PCR system. The melting temperature (T_m_) is calculated from the protein melt curve, and changes in T_m_ (known as ΔT_m_) are correlated with changes in protein stability and ligand binding [62]. Recombinant IRAP protein, in the presence of DMSO vehicle, provided a clear, single derivative peak in the TSA assay with a T_m_ of 64.87 ± 0.09 °C. Both **HA08** and **I** stabilized the thermal melting temperature of IRAP in a dose-dependent and saturable manner (Appendix A). **HA08** increased the IRAP T_m_ to 74.2 ± 0.3 °C at 200 µM (ΔT_m_ of 9.33 ± 0.31 °C), and **I** increased the T_m_ to 69.05 ± 0.17 °C at 400 µM (ΔT_m_ of 3.60 ± 0.17 °C). This indicates that the assay is fit for the purpose of being used to identify compounds with a similar binding mode to these two reference molecules.

### 2.2. Screening Campaign and Hit Triaging

To identify and triage the most promising inhibitors of IRAP [63], a stepwise screening assay cascade was established consisting of a primary screen using the fluorescence assay followed by active confirmation and dose-response curve analysis in both the fluorescence and the orthogonal absorbance-based biochemical assays (Figure 1 and Appendix A for a more detailed screening funnel with decision points). Confirmed actives were counter-screened in the fluorescence assay with a high concentration of Zn^2+^ added to the buffer to identify and de-prioritize undesirable Zn^2+^ chelating compounds and a biophysical thermal shift assay for the assessment of target engagement.

#### 2.2.1. Primary Screen (Plate Statistics, Active Selection Criteria, Active Rate, NN, and Bayesian Active List Enrichment)

A primary screening campaign was performed whereby the ~400,000 compounds from the European Lead Factory (ELF) were tested at 10 µM, *n* = 1, in the primary fluorescence IRAP enzymatic assay in a 1536-well plate format over a total of 4 days. The Z′- and S/B values were calculated for each plate using the intraplate, maximum (no protein), and minimum effect (DMSO vehicle only) controls, and any plates returning a Z′ < 0.6 and/or S/B < 3 were marked as failed and retested on the final screening day (Figure 2A). The Z-score and % effect for each compound were calculated using equations (see Appendix A), and primary actives were selected based on Z-score ≤ −4 (Figure 2B). A total of 3107 compounds returned a Z-score score < −4 and were flagged as primary hits. The structural fingerprints of these hits were used to build a Bayesian model [61] and identify substructural features that might be associated with biological activity. The top 500 compounds predicted to be active by the Bayesian model comprised 230 compounds that already had a Z-score ≤ −4 and 270 with a Z-score > −4 in the primary screen. To minimize the risk of false negatives, the 3107 actives from the primary screen were combined with the 270 additional Bayesian hits to provide a total of 3377 compounds to progress for active confirmation.

#### 2.2.2. Active Confirmation in Primary and Orthogonal Assays (Plate Statistics, Active Selection Criteria, Confirmation Rate)

Of the 3377 primary actives, enough material was available to cherry-pick 3374 compounds for re-testing in the primary assay. The activity of 1577 compounds was confirmed based on a Z-score ≤ −4, of which 630 compounds showed ≥ 30% effect (Figure 2C and Appendix A). None of the 270 compounds added from the Bayesian model showed a ≥30% inhibitory effect. A total of 1233 compounds with an inhibitory effect ≥30% in the primary or confirmation assay were tested in the orthogonal absorbance IRAP assay, with 281 showing an inhibition of ≥30% (Figure 2D and Appendix A). To avoid the prospect of false negatives being de-selected in the orthogonal assay, compounds showing either a ≥50% effect in the IRAP fluorescence confirmation assay (95 compounds) or ≥80% effect in primary fluorescence IRAP assay (10 compounds) were progressed alongside the positive confirmed inhibitors for further testing. This yielded 386 compounds for potency testing as serial dilutions.

#### 2.2.3. Dose–Response Curve (DRC) Analysis

Of the 386 compounds, 4 were out of stock, likely because of being frequent hitters. The remaining 382 compounds were tested in concentration–response curves in the primary IRAP fluorescence assay, the orthogonal IRAP absorbance assay, and the IRAP fluorescence assay in the presence of 10 μM Zinc (as deselection assay) to identify and de-prioritize compounds that may cause inhibition exclusively through a zinc chelating mechanism of action. All three assays were validated using three reference compounds (Figure 3A). The performance of the dose–response curve (DRC) analysis of the remaining confirmed actives was monitored for each assay (Figure 3B). DRC analysis resulted in 219 compounds with pIEC50 ≥ 4.7 in the primary and orthogonal assays (Figure 3C), and the inhibitory activity of 4 compounds was lost in the zinc deselection assay (Figure 3C,D).

#### 2.2.4. Biophysical Target Engagement Assay

The 382 compounds tested in concentration–response curves using the primary, orthogonal, and deselection assays were also tested in the IRAP thermal shift assay (TSA). Most of the compounds either had no effect on the T_m_ of IRAP or caused a positive stabilization, suggesting target engagement. The compounds were tested in two independent experiments. A total of 182 compounds caused a significant increase (>3 × SD) in the ΔT_m_ of IRAP in both replicates, whilst 3 compounds produced a significant decrease (<3 × SD) in the ΔT_m_ of IRAP in both replicates (Figure 4).

#### 2.2.5. Hit Prioritization

To progress the triage, compounds that demonstrated a pIC50 ≥ 4.7 in the IRAP fluorescence assay and/or demonstrated a significant response in the TSA were combined with structural analogs and proposed as the preliminary hit list. This list of 337 compounds was subjected to review by members of the medicinal chemistry team. Compounds were prioritized based on activity in the biochemical and biophysical assays, quantitative estimate of drug-likeness (QED [64]), and structural diversity. A total of 100 compounds were selected for LCMS analysis, of which 18 compounds failed the required criteria. To satisfy the business rules of the ELF, this list was reduced to 55 by further prioritizing the above criteria (activity in biochemical and biophysical assays, diversity) but also physicochemical properties recognizing that brain exposure will be important for this program. Following compound clearance, a QHL of 50 compounds was generated and the most potent compounds with IC_50_ below 3.5 µM are shown in Figure 5 and Table 1. The remaining compound structures and related data are given in Appendix A.

### 2.3. Resynthesis of the Most Potent Hit

**QHL1**, 2-(2-((3,4-dihydroisoquinolin-2(1*H*)-yl)sulfonyl)phenyl)-*N*-hydroxyacetamide. To a stirred solution of butyl 2-(2-((3,4-dihydroisoquinolin-2(1*H*)-yl)sulfonyl)phenyl)acetate (144 mg, 0.372 mmol) in a 1:1 mixture of THF:MeOH (3 mL) at 0 °C, a large excess of 50% aq. of hydroxylamine (40 equiv.) was added followed by a 4 M aq. KOH (3 equiv.). The resulting mixture was allowed to warm to room temperature and stirred overnight. The reaction mixture was concentrated to one-third of the original volume, and 10% aq. citric acid was added until precipitation occurred. The precipitate was recrystallized from hot EtOAc to yield the title product as a white solid (60 mg, 47% yield). ^1^H NMR (400 MHz, DMSO-d6) δ 10.59 (s, 1H), 8.83 (s, 1H), 7.89 (dd, *J* = 7.9, 1.4 Hz, 1H), 7.67–7.61 (m, 1H), 7.52–7.42 (m, 2H), 7.15 (m, 4H), 4.30 (s, 2H), 3.79 (s, 2H), 3.43 (t, *J* = 5.9 Hz, 2H), 2.85 (t, *J* = 5.9 Hz, 2H). ^13^C NMR (101 MHz, DMSO-d6) δ 166.37, 136.18, 135.00, 133.26, 133.20, 132.86, 131.94, 129.47, 128.82, 127.26, 126.59, 126.30, 126.11, 46.16, 42.67, 35.97, 28.10. LCMS (ESI+) *m*/*z* 347.0 [M+H]^+^; (280 nm); HRMS (ESI+) *m*/*z* [M = C_17_H_18_N_2_O_4_S]: [M+Na]_+_ calc’d 369.0885, found 369.0887.

## 3. Discussion

There is a particular need to identify novel IRAP inhibitors with the potential for development as brain-penetrant therapeutics, and high-throughput screening (HTS) of large and diverse compound libraries is a proven methodology to identify novel starting points for hit discovery. To discover novel small molecule inhibitors of IRAP, we developed, optimized, and miniaturized a series of biochemical and biophysical assays for IRAP, validated them using existing small molecule reference inhibitors, and then conducted a high-throughput screening campaign of the European Lead Factory library of ~400,000 compounds [59,65]. Designing and successfully executing an HTS campaign requires knowledge of the target coupled with a series of assays amenable to high-density microplate formats, e.g., 384- or 1536-well plates that enable low reagent consumption. These assays are required to be sensitive enough to detect compounds with desirable modes of action and provide a robust, stable, and reproducible signal that maximizes the likelihood of distinguishing genuine actives from noise when each compound can only be tested *n* = 1 during the primary screening campaign. To provide this, we developed primary and orthogonal biochemical assays of IRAP function based on non-natural substrates of L-Leucine conjugated via an amide bond to the fluorophore 7-amino-4-methylcoumarin (AMC) and the chromophore para-nitroanilide, respectively. Upon IRAP-mediated hydrolysis of the peptide bond, the fluorophore/chromophore is released and unquenched, allowing for product formation and reaction kinetics to be monitored in real time. These substrates provide very sensitive labels to measure IRAP activity, which in turn allows for low reagent usage, e.g., nM concentration of enzyme and substrate concentrations at or close to their experimentally determined Michaelis–Menten constant (*K*_m_). This provides assays that are not biased in sensitivity to any one particular mechanism of inhibition [66] while being optimally miniaturized for large-scale small-molecule screening without sacrificing dynamic range and robustness. Interestingly, most hits are more active in the fluorescence assay with a median difference in pIC_50_ between the fluorescence assay and absorbance assay of 0.23. This would be consistent with compounds exerting their activity through a substrate competitive mechanism of action because, whilst the substrate concentration in the fluorescence assay is equal to *K*_m_, a substrate concentration of 2.6 times *K*_m_ was required to obtain an appropriate signal window and robustness in the absorbance assay. This difference in substrate concentration relative to *K*_m_ should yield a theoretical pIC_50_ difference between the two assays of 0.26 for purely competitive inhibitors.

Both biochemical assays use labels that rely on similar wavelengths of 355 nm excitation for the fluorescence assay and 405 nm for the absorbance assay. There is therefore a strong possibility that compounds appearing as inhibitors by absorbing/quenching the excitation wavelength in the fluorescent assay will also affect the absorbance assay. These compounds will, however, be eliminated during the screen as their effect in the absorbance assay would be to enhance the signal, rather than reducing it as a genuine IRAP inhibitor would. One downside of using these label-based assays is the potential to miss inhibitors of native substrate binding, which bind remotely to the limited active site occupied by the relatively small synthetic substrates. This is an issue that modern high-throughput mass spec techniques attempt to address.

The inclusion of TSA in the screening cascade provides a genuinely orthogonal method for assessing hit binding to IRAP. This technique is used extensively in drug discovery either for primary screening or to confirm target engagement in secondary screening [67]. The relative simplicity of TSA and the fact it is amenable to low-volume 384-well microplates make it an ideal platform for hit triage in the ELF because of the low volumes of compounds that are available for re-supply. Validation that the assay was fit for purpose came from the reference inhibitors **HA08** and **I** causing very large shifts in the thermal stability of IRAP, with **HA08** being particularly effective. **HA08** is still one of the most potent inhibitors of the IRAP enzyme to date. Nearly half of the confirmed hits caused a statistically significant positive stabilization of IRAP, and three compounds produced a negative stabilization, indicating that many of the primary hits are likely to be genuine binders of IRAP. Gaining this confidence in the output from HTS is very important as there are many spurious mechanisms by which compounds can affect target activity that are unrelated to valid target engagement, as reviewed extensively elsewhere [63]. That said, while genuine binders to a protein should cause positive thermal stabilization [67], it is best to avoid a strict hit de-selection policy because, due to the lower free energy of the protein–ligand complex, this is not always the case. There are various reports of compound screens that use multiple orthogonal biophysical assays, including the TSA, where there is low overlap in the confirmation among techniques [68,69,70], even for validated binders. This may be because of gross or, indeed, subtle changes among the various assays with different buffer compositions, liquid handling techniques, plasticware, etc. We attempted to keep the TSA buffer as similar to the primary and orthogonal biochemical assays as possible, although we needed to remove detergent as it increases the fluorescence of the thermal shift dye. It also requires a significantly higher IRAP concentration.

For all HTS campaigns, there is a limited amount of resources for follow-up, so hit selection needs to consider the most promising for re-synthesis and further biological characterization. A unique feature of the ELF project is that contractually, this selection must be no more than 50 compounds, forming what is known as the qualified hit list (QHL). Once the selection of the QHL has been finalized, there is no possibility of re-selecting compounds that did not make the QHL at a later date. It is therefore important to consider all available data and evaluate the drug-likeness [64] and tractability of the hit structures. One feature that was common to many of the hits was the presence of a hydroxamic acid moiety. This was not unexpected as IRAP is a Zn^2+^-dependent aminopeptidase, and hydroxamic acids are well-known Zn^2+^ chelating agents. It is highly likely that the hydroxamic acid in these molecules is a strong driver of molecular recognition, with the oxygen and nitrogen atoms forming a coordination complex with the active site Zn^2+^. At their most extreme, however, hydroxamic acids may be able to strip metal ions from proteins, which would result in functional inhibition but would be an unproductive mechanism of action [71]. To identify and de-select compounds that might be acting in this manner, we re-screened the hits in the fluorescence assay in the presence of an equimolar concentration of Zn^2+^. This did not affect the potency of most compounds, including the majority of hydroxamic acids, which, combined with the positive thermal shift data, indicates specific binding to IRAP (Figure 5 and Table 1). It also implies that molecular recognition between the inhibitor and enzyme is not entirely reliant on zinc binding, which will be key for further optimizing the compounds.

The prioritization of hits for further studies is a key next step. As part of the wider project, ELF has now screened these QHL compounds many times. These data are available to assist in the prioritization process. The majority of the compounds show activity only in a small percentage of the total number of screens completed and an even smaller percentage when the target class is restricted to enzymes (Appendix A). This encourages the view that these compounds do not act through an indiscriminate mode of action. The activity of the most potent inhibitor, QHL1, has been confirmed by re-synthesis.

CNS penetration will be an important property of lead compounds for this program, so ideally, any prioritized hits should be predicted to either possess this property or at least not require significant modification to achieve it. As a first step, drug-likeness was considered a desirable parameter in this early-stage drug discovery program to aid compound ranking and enhance the likelihood of selecting orally bioavailable candidates. The concept of the quantitative estimate of drug-likeness (QED) [64] was valuable in this regard, as it comprehensively considers multiple molecular properties including lipophilicity, molecular mass, counts of hydrogen bond donors and acceptors, polar surface area, number of rotatable bonds, and the presence of structural alerts, amongst others. QED values range from zero (unfavorable properties) to one (favorable properties), and compound scores close to or above 0.65 were considered promising, which also indicates a higher probability of suitable ADMET properties (Table 1). Ultimately, alongside drug-likeness scoring, robust inhibitory potency is essential for discerning the most promising hits.

Overall, despite the dominance of hydroxamic acids, a range of chemotypes with significant biochemical activity and evidence of target engagement were discovered, which are suitable for further exploration. In addition, half of the QHL compounds in Table 1 have promising QED values of ≥0.65, indicating that high drug-likeness, i.e., high probability of absorption and bioavailability [72], is in reach through further development. It is considered likely that the hydroxamic acids include catalytic Zn in their binding site and hence, will be competitive with a substrate, as suggested by some of the early biochemical results. More detailed characterization to confirm this will be carried out in due course. It is interesting to note that the biggest shifts observed in the TSA generally occur for hydroxamic acids. Based on these data, the highest priority compounds are likely to be those that demonstrate good activity in the biochemical and biophysical assays whilst restraining the physio-chemical properties. Whilst this clearly points in the direction of some of the hydroxamic acids, e.g., QHL1, a strength of the ELF is the diversity of the structures available, and it is interesting to note that good lead-like properties and activities are achievable both in the biochemical and biophysical assays in the absence of the hydroxamic acid warhead in more than 22 non-hydroxamic acid compounds with IC50 below 3.5 µM (see Figure 5 and Table 1).

## 4. Materials and Methods

### 4.1. Reference Compounds and Assay Substrates

For the reference compounds, 1 mM **IVDE77** [38] stock solution was provided in 100% DMSO (Fisher Scientific D/4125/PB17). Then, 10 mM **HA08** [39] was prepared by dissolving 1.3 mg **HA08** (Mw 560.176 g/mol) in 232 μL DMSO (Fisher Scientific D/4125/PB17). The solution was aliquoted and stored at −20 °C. Next, 100 mM **I** [45] was prepared by dissolving 21.4 mg **I** (Mw 402.946 g/mol) in 531 μL DMSO (Fisher Scientific D/4125/PB17). The reference structures are shown in Appendix A. The solution was aliquoted and stored at −20 °C. For the primary assay substrate, 100 mM L-Leucine-7-amido-4-methylcoumarin was prepared by dissolving 100 mg L-Leucine-7-amido-4-methylcoumarin (Mw 324.8 g/mol) (Sigma L2145-100MG) in 3.1 mL DMSO. The solution was aliquoted and stored at −20 °C. For the orthogonal assay substrate, 100 mM L-leucine-para-nitroanilide was prepared by dissolving 1 g (Mw 251.28 g/mol) (Sigma L9125-1G) in 26.5 mL methanol (Sigma 322415-2L). The solution was aliquoted and stored at −20 °C. For the TSA assays, Thermal shift dye kits were obtained from Thermo Scientific, and DMSO, Trizma HCL, and sodium chloride were from Sigma Aldrich, St. Louis, MO, USA. hSLAP purified protein [51] was provided by Prof. Efstratios Stratikos. **HA08** and **I** (see Appendix A) were provided by the Department of Medicinal Chemistry, Uppsala University, and prepared to a 10 mM stock in DMSO. MicroAmp Endura plate optical qPCR plates were from Thermo Scientific, Waltham, MA, USA, and Clear PCR-compatible plate seals were from Greiner, Kremsmünster, Austria.

### 4.2. Assay Buffers and Reagents

To prepare the assay buffer (50 mM Tris-HCl pH 7.4, 150 mM NaCl, 0.1 mM PMSF, 0.05% BSA) of the primary and orthogonal assays, 1 M Tris-HCl was prepared by dissolving 60.57 g Trizma-base (Mw 121.14 g/mol) (Sigma 93362-250G) in 500 mL MQ water, and the pH was adjusted to 7.4 using 4 M HCl. The solution was filtered with a 0.45 μm filter and stored at room temperature (RT). Then, 2 M NaCl stock was prepared by dissolving 29.22 g NaCl (Mw 58.44 g/mol) (Sigma S7653-250G) in 250 mL MQ water. The solution was filtered with a 0.45 μm filter and stored at RT. Next, 100 mM Phenylmethanesulfonylfluoride (PMSF) stock was prepared by dissolving 1 g PMSF (Mw 174.19 g/mol) (Sigma 78830-1G) in 57.4 mL isopropanol (Sigma I9516-500ML) and stored at 4 °C. A 10% Bovine Serum Albumin (BSA) stock solution was prepared by dissolving 2 g BSA (Mw 66.5 kDa) (Sigma A7030-50G), which was dissolved in 20 mL MQ water. The solution was filtered with a 0.45 μm filter, aliquoted, and stored at −20 °C. For the deselection assay buffer (50 mM Tris-HCl pH 7.4, 150 mM NaCl, 0.1 mM PMSF, 0.05% BSA, 10 μM ZnSO_4_), 1 mM ZnSO_4_ solution was prepared by dissolving 28.8 mg ZnSO_4_ (Mw 287.56 g/mol) (Sigma Z0251-100G) in 100 mL MQ water. The solution was filtered with a 0.45 μm filter and stored at RT. For the TSA assay buffer (50 mM Tris HCL, 150 mM NaCl, pH 7.4), stocks of 1 M Tris HCL and 2 M NaCl were prepared as described above.

### 4.3. Assay Protocols

#### 4.3.1. Primary Fluorescence IRAP Assay

The primary assay was performed in a black non-binding 1536-well plate (Corning 3728) and sealed with universal lids (Corning 3098). First, 10 nL of test compounds (4 mM) or 100% DMSO was dispensed into the 1536-well assay plates using an Echo acoustic liquid handler (Beckman Coulter, Brea, CA, USA). Then, 2 μL of protein working solution (5 × 10^−4^ mg/mL) was dispensed to the negative control (min effect) and compound wells (final well concentration 2.5 × 10^−4^ mg/mL) using a Certus Flex liquid dispenser (Fritz Gyger AG, Thun, Switzerland) and 2 μL assay buffer to the positive control wells (max effect) using a FLEXdrop (PerkinElmer, Waltham, MA, USA). Next, 2 μL of 75 μM substrate working solution was dispensed into all wells using the FLEXdrop (final well concentration 37.5 μM). The assay plates were then centrifuged at 187.5× *g* for 30 s and incubated in the dark at RT for 60 min and finally read on an Envision plate reader (PerkinElmer) (Ex. 355 nm/Em. 460 nm, LANCE/DELFIA mirror, 3 flashes per second). During the dose–response curve analysis in the primary and deselection assay setup, 40 nL of compound (range 2.00 × 10^−3^ M–2.74 × 10^−6^ M) or 100% DMSO was dispensed into the assay plates using an Echo acoustic liquid handler (Beckman Coulter) with a final well concentration range of 20 μM–27.4 nM compound and 1% DMSO.

#### 4.3.2. Orthogonal Absorbance IRAP Assay

The orthogonal assay was performed in a black non-binding 1536-well plate (Corning 3728) and sealed with universal lids (Corning 3098). First, 20 nL of test compounds (4 mM) or 100% DMSO was dispensed into the 1536-well assay plates using an Echo acoustic liquid handler (Beckman Coulter). Then, 4 μL of protein working solution (5 × 10^−4^ mg/mL) was dispensed to the negative control (min effect) and compound wells (final well concentration 2.5 × 10^−4^ mg/mL) using the Certus Flex liquid dispenser (Fritz Gyger AG) and 4 μL assay buffer to the positive control wells (max effect) using the FLEXdrop (PerkinElmer). Next, 4 μL of 2 mM substrate working solution was dispensed into all wells using the FLEXdrop (final well concentration 1 mM). The assay plates were then centrifuged at 187.5× *g* for 30 s, incubated in the dark at RT for 120 min, and finally read on an Envision plate reader (PerkinElmer) (absorbance at 405 nm, 10 flashes per second). During the dose–response curve analysis in the orthogonal assay, 80 nL of compound (range 2.00 × 10^−3^ M–2.74 × 10^−6^ M) or 100% DMSO was dispensed into the assay plates using an Echo acoustic liquid handler (Beckman Coulter) with final well concentration range of 20 μM–27.4 nM compound and 1% DMSO.

#### 4.3.3. Deselection Fluorescence IRAP Assay

The deselection assay was performed following the protocol for the primary fluorescence IRAP assay (see Section 4.3.1), except that the deselection assay buffer was used instead of the assay buffer (see Section 4.2), which resulted in a final zinc concentration of 10 µM.

#### 4.3.4. Thermal Shift (TSA) IRAP Assay

For the TSA assay, vehicle (DMSO) or compound was dispensed into MicroAmp Endura optical qPCR plates at the desired volume using an Echo acoustic liquid handler. Reference ligands were tested as a 10-point half-log concentration series, starting at 200 µM for **HA08** and 400 µM for **I**. Screening hits were tested at 30 µM with a DMSO concentration of 1% (*v*/*v*). Thermal shift dye was diluted in assay buffer containing 1 µg/well protein (50 mM Tris HCL, 150 mM NaCl, pH 7.4) to a 1X final assay concentration, and 20 µL was added per well to the vehicle/compound containing plate. The plates were then centrifuged for 10 s at 1000 rpm and sealed with Clear PCR-compatible plate seals. For the TSA, the assay plates were held at 25 °C for 1 min before heat was applied at a rate of 0.05 °C/second to a final temperature of 99 °C, which was then held for one minute. Fluorescence was measured throughout the experiment with filter settings of 580 nm/623 nm (Ex/Em).

## 5. Conclusions

In conclusion, we successfully employed a high-throughput screening approach to identify inhibitors of insulin-regulated aminopeptidase (IRAP) with the potential to serve as new starting points in our long-term ambition to develop small-molecule drug-like therapeutics. Utilizing a diverse compound library of 400,000 drug-like compounds and optimized biochemical and biophysical assays, we identified hits through a stringent selection process, leading to the creation of a qualified hit list of 50 compounds. Initiation with a primary screening with a 0.9% hit rate yielded 3107 compounds. By leveraging a Bayesian model, we augmented the hit list to identify structurally similar compounds within the library that may have been false negatives, subsequently confirming 37% or 1233 compounds as active upon rescreening. Notably, compounds identified from the Bayesian model did not feature among the active compounds. Advancing further, we subjected the hits to an orthogonal assay, resulting in the retention of 281 compounds. From these three assay results, a subset of the most potent compounds identified so far, totaling 386 compounds, underwent comprehensive dose–response evaluation across the primary, orthogonal, and deselection assays, complemented with biophysical target engagement assessment by thermal shift assay evaluation. As IRAP is a zinc-dependent enzyme, we included a deselection assay with additional zinc to discard chelators i.e., to exclude false hits. Further orthogonal assays, such as the thermal shift assay, strengthened our confidence in the hit selection, confirming specific binding to IRAP. In the curation of the final selection comprising the 50 most promising inhibitors, an array of criteria was meticulously assessed. These criteria included evaluations derived from biochemical and biophysical assays, as well as assessments of structural diversity and drug-likeness. The overarching goal was to identify the most valuable compounds demonstrating optimal potency, selectivity, and high chance for oral bioavailability (QED). Despite the prevalence of hydroxamic acids, various chemotypes with significant biochemical activity and evidence of target engagement were discovered. Notably, the inhibitory activities of the selected hydroxamic acid-bearing compounds in the QHL were not affected at higher zinc concentrations. Further characterization and optimization of lead compounds, including assessment of central nervous system penetration, are underway and will be crucial for advancing potential candidates toward therapeutic development as cognitive enhancers. In total, 20 new compound classes were identified with IC_50_ below 3.5 µM, which has opened up numerous opportunities for the development of unique drug-like IRAP inhibitors. Moreover, accumulating evidence demonstrates that the pharmacological inhibition of IRAP may hold promise as a valuable approach not only for the treatment of memory disorders and neurodegenerative diseases but also potentially for neuroprotection in connection with ischemic stroke and other cardiovascular disorders.

## Figures and Tables

**Figure 1 ijms-25-04084-f001:**
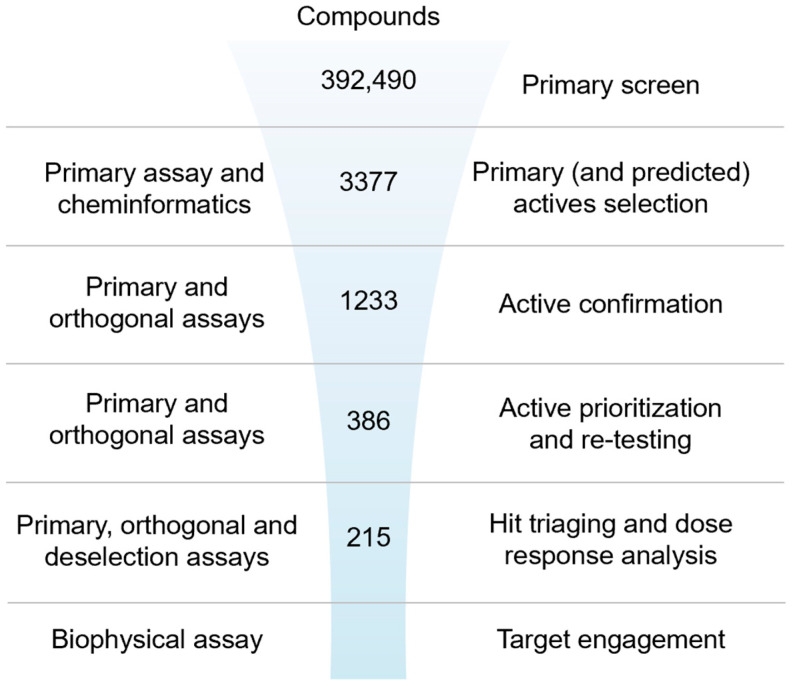
Ultra-high-throughput hit identification and triaging cascade for IRAP.

**Figure 2 ijms-25-04084-f002:**
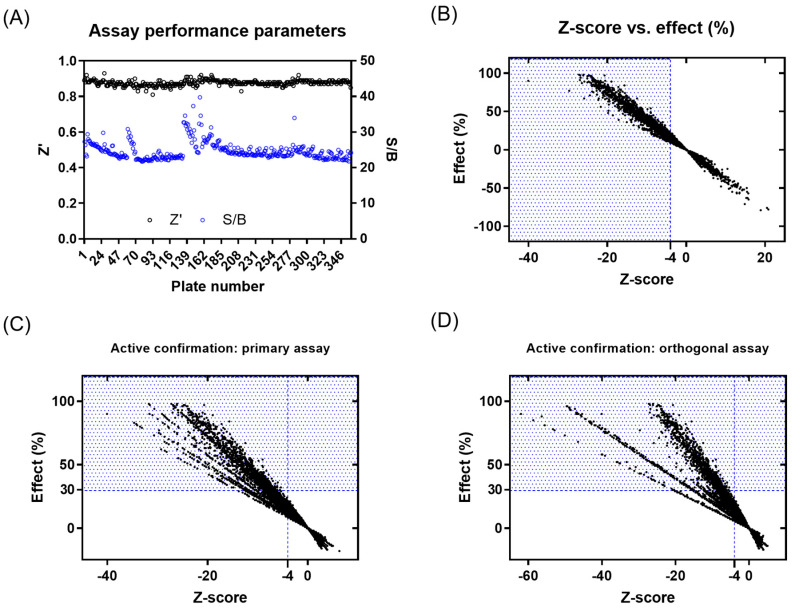
Results from primary screening and active confirmation analysis. (**A**) Assay performance parameters (Z′ and S/B) per plate. (**B**) Primary active compound list selection based on Z-score and %effect. (**C**) Primary active confirmation in primary assay with fluorescent readout based on Z-score and %effect. (**D**) Primary active confirmation in orthogonal assay with absorbance readout based on Z-score and %effect. Compounds in the blue dotted area were considered actives.

**Figure 3 ijms-25-04084-f003:**
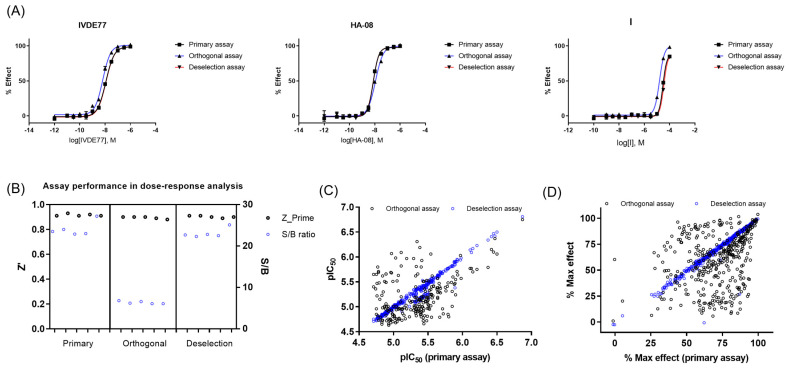
Dose–response curve (DRC) analysis in the primary, orthogonal, and deselection assays. (**A**) DRC of reference compounds in the primary, orthogonal, and deselection assays. (**B**) Assay performance during DRC analysis of confirmed actives in the primary, orthogonal, and deselection assays. (**C**) pIC_50_ values in the primary, orthogonal, and deselection assays. (**D**) Max %effect in the primary, orthogonal, and deselection assays.

**Figure 4 ijms-25-04084-f004:**
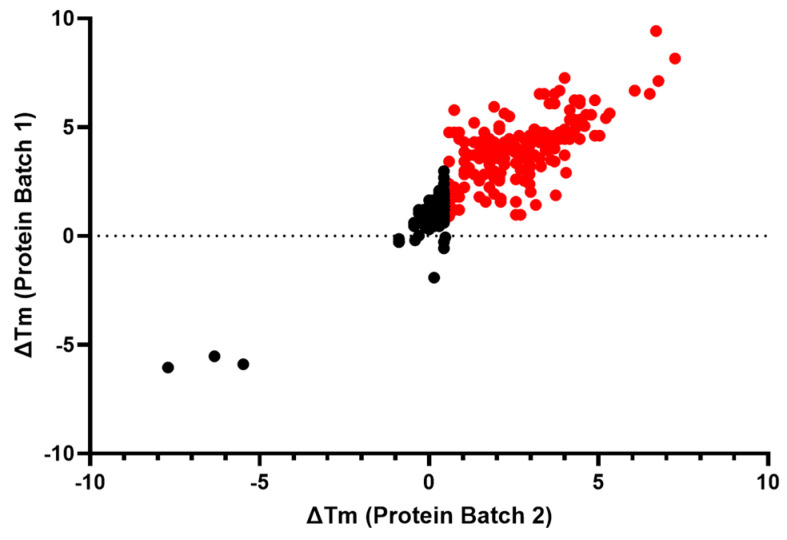
Assessment of target engagement by thermal shift assay (TSA) screening. Thermal shift data from the remaining confirmed 382 active compounds. All compounds colored in red caused a significant increase (>3 × SD) in the ΔT_m_ of IRAP in both replicates.

**Figure 5 ijms-25-04084-f005:**
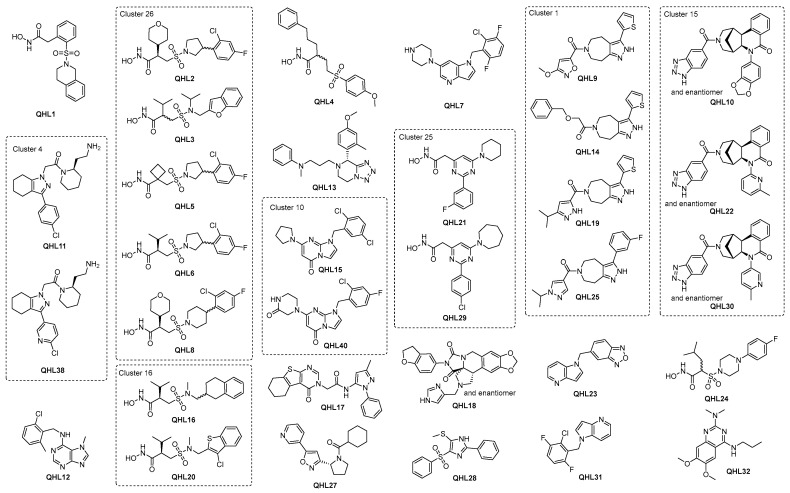
Top QHL compounds with IC_50_ below 3.5 µM and additional compounds in the same cluster.

**Table 1 ijms-25-04084-t001:** Screening results for QHL compounds with IC_50_ below 3.5 µM and additional compounds in the same cluster.

QHL	Cluster	Fluorescence IC_50_ (µM)	Fluorescence pIC_50_	Absorbance pIC_50_	Fluorescence + Zn pIC_50_	TSA (∆°C)	Mw (g/mol)	HBA/HBD	TPSA (Å^2^)	LogD	LipE	QED
**1**	NA	0.32	6.50	6.06	6.50	6.77	346.40	6/2	95.09	1.65	4.85	0.65
**2**	26	0.35	6.45	6.10	6.47	3.70	434.91	7/2	104.32	1.56	4.89	0.52
**3**	26	0.38	6.42	6.15	6.41	5.22	382.47	7/2	108.22	2.34	4.08	0.54
**4**	NA	0.71	6.15	5.70	6.17	4.00	391.48	6/2	101.08	3.64	2.51	0.48
**5**	26	1.20	5.92	5.71	5.95	4.29	390.86	6/2	95.09	2.13	3.79	0.59
**6**	26	1.26	5.90	5.24	5.91	3.55	392.87	6/2	95.09	2.33	3.57	0.57
**7**	NA	1.29	5.89	<4.70	5.86	0.30	362.80	4/1	33.09	3.79	2.10	0.72
**8**	26	1.32	5.88	5.45	5.89	4.61	448.94	7/2	104.32	1.94	3.94	0.51
**9**	1	1.51	5.82	<4.70	5.80	0.15	344.39	7/1	112.49	2.84	2.98	0.79
**10**	15	1.58	5.80	4.80	5.78	0.30	493.51	9/1	100.65	3.27	2.53	0.46
**11**	4	1.66	5.78	5.41	5.77	1.92	400.94	5/2	64.15	3.21	2.57	0.83
**12**	NA	1.91	5.72	4.86	5.71	−0.15	287.75	5/1	55.63	4.04	1.68	0.80
**13**	NA	2.14	5.67	4.79	5.70	−1.91	392.50	7/0	59.31	3.28	2.39	0.61
**14**	1	2.24	5.65	<4.70	5.63	0.15	367.46	5/1	86.46	3.23	2.42	0.75
**15**	10	2.40	5.62	<4.70	5.59	−0.89	363.24	5/0	39.15	2.08	3.54	0.83
**16**	16	2.63	5.58	5.42	5.55	2.07	368.49	6/2	95.09	2.44	3.14	0.57
**17**	NA	2.63	5.58	<4.70	5.49	0.30	419.50	7/1	107.82	3.39	2.19	0.70
**18**	NA	2.82	5.55	5.25	5.57	1.92	485.49	10/1	100.23	2.53	3.02	0.56
**19**	1	2.82	5.55	4.79	5.53	0.30	355.46	6/2	105.91	3.72	1.83	0.75
**20**	16	2.82	5.55	5.24	5.5	4.15	404.93	6/2	123.33	2.84	2.71	0.54
**21**	25	2.95	5.53	5.25	5.55	2.66	330.36	6/2	78.35	2.96	2.57	0.67
**22**	15	3.09	5.51	<4.70	5.52	0.15	464.52	8/1	95.08	3.17	2.34	0.49
**23**	NA	3.09	5.51	<4.70	5.49	0.44	250.26	5/0	56.74	2.90	2.61	0.55
**24**	NA	3.09	5.51	5.52	5.26	4.00	387.47	7/2	98.33	1.94	3.57	0.55
**25**	1	3.16	5.50	<4.70	5.46	0.00	367.42	6/1	66.81	3.20	2.30	0.77
**26**	NA	3.24	5.49	5.11	5.45	4.44	428.52	7/2	104.32	2.58	2.91	0.51
**27**	NA	3.24	5.49	4.97	5.29	1.48	325.40	5/0	59.23	3.35	2.14	0.87
**28**	NA	3.24	5.49	4.81	5.25	0.15	330.42	4/1	96.50	3.95	1.54	0.73
**29**	25	3.31	5.48	5.53	5.49	1.92	360.84	6/2	78.35	3.88	1.60	0.64
**30**	15	3.47	5.46	<4.70	5.41	0.30	464.52	8/1	95.08	3.03	2.43	0.49
**31**	NA	3.47	5.46	<4.70	5.38	0.00	278.68	2/0	17.82	4.41	1.05	0.64
**32**	NA	3.47	5.46	<4.70	5.21	0.15	290.36	6/1	59.51	2.36	3.10	0.88
**38**	4	4.68	5.33	4.96	5.34	2.22	401.93	6/2	77.04	2.27	3.06	0.78
**40**	10	4.68	5.33	<4.70	5.28	0.30	375.78	7/1	68.25	0.95	4.38	0.85

NA: not applicable (singletons); TSA: thermal shift assay; HBA/HBD: number hydrogen bond accepting or donating groups; TPSA: topological polar surface area, LipE: calculated from (pIC_50Fluorescence_ − LogD); QED [64]: quantitative estimate of drug-likeness.

## Data Availability

Data are contained within this article and Appendix A.

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
