# Peer review of "The Discovery of New Inhibitors of Insulin-Regulated Aminopeptidase by a High-Throughput Screening of 400,000 Drug-like Compounds"

_ijms, 2024, doi:10.3390/ijms25074084_

Round 1

Reviewer 1 Report

Comments and Suggestions for Authors

The manuscript ijms-2920019 demonstrates the screening of new inhibitors of insulin regulated amino peptidase and in my opinion can be published in International Journal of Molecular Sciences. However, I think that it would be more reasonable to publish the manuscript in the journal “pharmaceutics” of the same publisher. Besides, the manuscript can be improved for reading. This will increase interest for a wider range of readers. Here are some suggestions:

1. Line 32: it is necessary to specify the compound.

2. It is necessary to note the advantages of the extensive experimental path chosen by the authors compared to the docking approach.

3. Figure 1S: Structures are not displayed.

4. It is not clear why 392490 compounds were selected? Based on what criteria were they chosen? How many compounds were sampled from?

5. Figure 1: I propose to supplement the figure by adding criteria for reducing the number of connections between stages.

6. Conclusion needs to be improved. Conclusion should show the essence of the article, in my opinion, and not be only a summary of what the authors did. It is necessary to carry out a methodological generalization of the results of the implementation of the extensive experimental path chosen by the authors, taking into account each stage in accordance with Figure 1.

Reviewer 2 Report

Comments and Suggestions for Authors

Presented manuscript have substantial reader friendly data. I am suggesting some minor revision.

1. Author should state the distinctiveness and novelty of presented work

2. Author should discuss about structural aspects of insulin regulated amino peptidase

3. Author should justify the practical applicability of presented high throughput screening

4. Authors should include the future perspective of presented work

Comments on the Quality of English Language

Minor editing of English language required

Reviewer 3 Report

Comments and Suggestions for Authors

The article entitled: The Discovery of New Inhibitors of Insulin Regulated Amino Peptidase  by A High-Throughput Screening of 400 000 Drug-like Compoundsis scientifically sound, and very interesting, it will fulfill the knowledge for high-throughput screening study.

Just have few remarks in this manuscript, are as below:

Remarks:

-   In the article entitled: The Discovery of New Inhibitors of Insulin Regulated Amino Peptidase  by A High-Throughput Screening of 400 000 Drug-like Compounds ### it will be better if you use comma as 400,000 instead of 400 000 and please correct all of the other values in the text i.e., Line 83 ….10 500 small moleculs….; Line 101 ….HTS of ~400 000…. And so on.

-   Line 183 2.2. Screning campaign and hit triaging ### should be Screening

-   I would suggest that the authors should add more information of “Insulin Regulated Amino Peptidase (IRAP)” and the benefits of “IRAP inhibitors” to clarify the reason of finding IRAP inhibitors by HTS.

-   I would say that this is a good one MS to publish in JIMS.

Cheers,

Date of this review

10 March 2024

Comments on the Quality of English Language

Minor editing of English language required, I think may be from typing error.

Reviewer 4 Report

Comments and Suggestions for Authors

The manuscript by Gising and colleagues is interesting and well performed. It leads to the identification of IRAP inhibitors though a HTS approach.

However, I think that at least the most active compound should be tested in in vitro assay and its BBB permeability should be determined, again with an in vitro test (for example PAMPA).

Also the potential cell toxicity should be determined.

I do not agree with the sentence at page 11, lines 392 and following.

Comments on the Quality of English Language

Minor editing of English language required

Round 2

Reviewer 4 Report

Comments and Suggestions for Authors

The manuscript is suitable for the publication in this revised version.